# KIR Receptors as Key Regulators of NK Cells Activity in Health and Disease

**DOI:** 10.3390/cells10071777

**Published:** 2021-07-14

**Authors:** Joanna Dębska-Zielkowska, Grażyna Moszkowska, Maciej Zieliński, Hanna Zielińska, Anna Dukat-Mazurek, Piotr Trzonkowski, Katarzyna Stefańska

**Affiliations:** 1Department of Medical Immunology, Medical University of Gdańsk, 80-210 Gdansk, Poland; gramos@gumed.edu.pl (G.M.); mzielinski@uck.gda.pl (M.Z.); hzielinska@gumed.edu.pl (H.Z.); adukat@uck.gda.pl (A.D.-M.); ptrzon@gumed.edu.pl (P.T.); 2Department of Obstetrics, Medical University of Gdańsk, 80-214 Gdansk, Poland; kciach@gumed.edu.pl

**Keywords:** KIR, NK, HLA, transplantation

## Abstract

Natural killer (NK) cells are part of the cellular immune response. They target mainly cancer and virally infected cells. To a high extent cytotoxic activity of NK cells is regulated inter alia by signals from killer immunoglobulin-like receptors (KIR). The major histocompatibility complex (MHC) class I molecules are important ligands for KIR receptors. Binding of ligands (such as MHC I) to the KIR receptors has the important role in solid organ or hematopoietic cell transplantation. Of note, the understanding of the relationship between KIR and MHC receptors may contribute to the improvement of transplant results. Donor-recipient matching, which also includes the KIR typing, may improve monitoring, individualize the treatment and allow for predicting possible effects after transplantation, such as the graft-versus-leukemia effect (GvL) or viral re-infection. There are also less evident implications of KIR/MHC matching, such as with pregnancy and cancer. In this review, we present the most relevant literature reports on the importance of the KIR/MHC relationship on NK cell activity and hematopoietic stem cell transplantation (HSCT)/solid organ transplantation (SOT) effects, the risk of allograft rejection, protection against post-transplant cytomegalovirus (CMV) infection, pregnancy complications, cancer and adoptive therapy with NK cells.

## 1. Introduction

The main role of the immune system cells is to protect the body against infections, tumor cells and a broad range of ‘danger’ signals. The additional face of this function is the response to allergens and transplanted allogeneic tissues. While the latter effects are undesirable, this is an unfortunate natural consequence of the proper functioning of the immune system.

Gaining knowledge on how the immune system works, notably how it distinguishes ‘self’ from ‘non-self,’ greatly improves diagnostics and treatment. For example, apparently the obvious application of human leukocyte antigen (HLA) polymorphisms in donor-recipient matching was once a revolutionary idea [1]. However, there are mechanisms for sensing foreign particles beyond HLA. Recently, there has been a growing interest in the mechanisms of NK cell activity and the involvement of these cells in allorecognition. Apart from cytokines and other soluble factors, the activity of NK cells can be modulated by the coupling of surface receptors with ligands on self and transplanted cells. Reports suggest the importance of interactions between KIR receptors and HLA in predicting the risk of allograft rejection, protection against post-transplant CMV infection, *graft-versus-host disease* (GVHD) and pregnancy complications [2]. A thorough understanding of these relationships would give rise to a new approach in current diagnostics and therapy.

## 2. NK (Natural Killer Cells)

NK cells are part of the innate immune response. These are mononuclear cells that are formed in the bone marrow from lymphoid precursors. Their function is to kill target cells defined as cells infected by viruses, such as cancer cells. They can also take part in the destruction of allogeneic tissues [3] and solid organs [4,5] without prior immunization. NK cells comprise 5–10% of the total peripheral blood lymphocytes count. The expression of the markers CD56 and CD16 molecules in the absence of lineage-specific markers for T cells CD3, B cells CD19 and monocytes CD14 creates the basic phenotype of these cells.

There are two main populations of NK cells in the body: CD56^bright^CD16^+/−^ and CD56^dim^CD16^+^ cells [6]. In the peripheral blood, CD56^dim^CD16^+^ phenotype dominates [7]. These cells are characterized by fast secretion of gamma interferon (IFN-γ) and strong cytotoxicity. CD56^bright^CD16^+/−^ NK cells secrete IFN-γ, granulocyte macrophage colony-stimulating factor (GM-CSF), tumor necrosis factor alpha (TNF-α) and are less mature and cytotoxic than CD56^dim^CD16^+^ NK cells [8,9,10]. They are present mainly in the liver where they comprise up to 30% of all lymphocytes [11]. Another organ enriched with NK cells is the endometrium in pregnancy. Uterine natural killer cells (uNK) comprise 70% of all lymphocytes in the uterus [12]. The following main mechanisms of NK cell activity have been described:Spontaneous cytotoxic reaction through release of perforin, granzyme B.Antibody dependent cytotoxicity (ADCC). NK cells express the CD16 receptor (FcγRIIIa), which binds Fc fragment of IgG class antibodies inducing NK cell activation and degranulation leading to destruction of the target cell.Receptor/ligand interactions via Fas/FasL, TNF/TNFR coupling and triggering of the apoptosis in the target cell.

In one of the first papers on the activity of NK cells, the authors Klas Kärre et al. [13] presented the “missing self” hypothesis, which stated that NK cells kill cells with weak or no MHC-I expression (Figure 1). According to this report, the expression of self MHC molecules protects cells against lysis by NK cells. In the context of the “missing self” hypothesis, Kärre and Ljunggren put forward the thesis that there are specific receptors on NK cells, which recognize MHC. These receptors turned out to be KIRs.

The “missing self” hypothesis is the most established, but there is not a single theory on NK cell function. The reactions between NK cell receptors and MHC create a huge spectrum of possibilities for the regulation of NK cell functions. NK cell education is about balancing the activating forces with the inhibitory sensitivity of NK cells. NK cells, which are strongly educated, are characterized by high binding strength and reactivity to the targets lacking self HLA molecules. On the other hand, they are highly sensitive to the expression of self HLA molecules and to self-HLA inhibitory signals. Uneducated cells respond weakly to the own HLA, which gives a poor response to both the “missing self” antigen and to the own inhibitory signals [14]. Overall, the interaction between inhibitory receptors and ligands regulates NK cell reactivity.

The most frequently described mechanism is the relationship of KIR receptors and MHC ligands, but it is known that NK education also applies to other receptor-ligand interactions, for example CD94/NKG2A and LIR family receptors that bind HLA and increase the potential of NK responses. Based on the available experiences, it can be concluded that the activity of NK cells depends on the level of education of these cells. There are studies that describe the association between the status of cell education and the effect on disease progression [14]. For example, NK cells in pregnancy educated by KIR2DL1 + HLA-C2 increased the risk of developing pre-eclampsia and low birth weight [15]. Similarly, uneducated cells KIR2DL2 + HLA-C2 were associated with increased susceptibility to type I diabetes [16].

Finally, reports on memory NK cells are of particular interest. Until now, the development of immune memory was characteristic only of T and B cells. It was thought that there was no specific response in the case of NK cells but the conclusions drawn from large amounts of experience refute the existing dogmas. Evidence has accumulated that, upon exposure of NK cells to hapten, specific viral antigens lead to NK cell memory and NK memory cells exhibit features of adaptive immune cells [17]. The first to report that murine NK cells have adaptive immune features to the administered haptens was provided by O’Leary et al., who identified hapten-specific memory in a subset of NK cells residing in the liver of mice lacking B and T lymphocytes [18]. In the following years, the evidence on the development of immune memory in NK cells has been accumulating [19]. For example, memory-like NK cell responses in viral and bacterial infections have been described recently [20]. Human NK cells can control CMV infection in the absence of T cells [21]. The development of immune memory is related to the greater cytotoxicity of memory NK cells upon repeated contact with the antigen [22]. In addition, the studies on the influence of changes in NK cell metabolism on their activity seem to be of great interest [23]. It has been known for many years that the use of certain immunosuppressants alters the metabolism of T cells and affects their activity and differentiation. It turns out that a similar effect occurs in the case of NK cells, where a change in the metabolic activity affects the differentiation and cytotoxic activity. The studies by Marçais et al. showed that, as compared to phenotypically mature cells, mTOR kinase activation, nutrients and glucose transport were lower in the less differentiated cells [24]. Indeed, mature NK cells are characterized by a high glucose-glycolysis metabolism, which translates into enhanced cytolytic functions [25]. Nevertheless, metabolic activity is not the only factor that influences the education of NK cells. Recent findings suggest that the NK cell pool may be regulated by epigenetic modification phenomena. However, the evidence is still scarce [26].

Apart from cytotoxic activity, NK cells are efficient producers of proinflammatory cytokines, such as IFN-γ and TNF-α, which allows them to promote the activation of inflammation, excessive inflammation or even autoimmunity [27]. The influence of NK cells on the excessive activation of the inflammasome, which is characteristic of autoinflammatory diseases, has been described. Hence, the attention has recently been focused on the potential involvement of NK cells in the pathogenesis of autoimmune [28] and autoinflammatory diseases [29].

Moreover, in light of new reports on NK cell recall/educated responses mediated with cytokines and seen as a kind of immune memory, the role of NK cells on autoreactive and autoinflammatory diseases is extremely interesting [30]. On the other hand, some NK cells have a regulatory function via secretion of anti-inflammatory cytokines, such as *transforming growth factor beta* (TGF-*β*) or IL-10. There are reports on the cytokine-modulated function of NK cells in hepatitis C virus (HCV) infection [31] and the role of IL-10 secreted by these cells in chronic HCV infection, such as the impact on HCV ribonucleic acid (RNA) level and the expression of HCV proteins [32].

## 3. KIR (Killer Immunoglobulin-Like Receptors)

KIRs are cell surface molecules expressed on NK cells and—to a lesser extent—on NKT cells. They were initially defined as inhibitory receptors and named ‘Killer-cell Inhibitory Receptors’. When activating receptors within this family were found, both groups were named ‘killer-cell immunoglobulin-like receptors’ (KIRs) [33]. *KIR* genes are located on chromosome 19, in the leukocyte receptor complex (LRC). Currently, the KIR gene family consists of 15 gene loci (KIR2DL1, KIR2DL2/L3, KIR2DL4, KIR2DL5A, KIR2DL5B, KIR3DL2, KIR3DL3, KIR2DS1, KIR3DL1/S1, KIR2DS2, KIR2DS3, KIR2DS4, KIR2DS5) and 2 pseudogenes (KIR2DP1 and KIR3DP1) (Figure 2).

The description of KIR genes in current nomenclature takes into account the number of Ig-like domains, which is ‘2D’ for two domains or ‘3D’ for three domains, and the length of the cytoplasmic tail, which is ‘S’ for Short or ‘L’ for long (Figure 3).

KIR genes exist as several allelic forms, which makes them the most polymorphic human family of NK cell receptors. In December 2020, the total number of known KIR alleles was 1110. The length of the cytoplasmic tail corresponds to the type of NK function mediated by particular KIR receptors. In general, activating receptors have a short cytoplasmic fragment with the immunoreceptor tyrosine-based activatory motif (ITAM), and are marked with the letter S (short). In contrast, inhibitory receptors have a long cytoplasmic fragment ending in an immunoreceptor tyrosine-based inhibitory motif (ITIM), and are marked with the letter L (long). Only the KIR2DL4 receptor can conduct both: activation and inhibitory signals. KIR genotypes can be divided into two haplotypes: A and B depending on the composition of their genes. Haplotype A contains only one activating receptor 2DS4 while haplotypes B contain different combinations of the activating genes KIR2DS1, KIR2DS2, KIR2DS3, KIR2DS5 and KIR3DS1 (Figure 2). Haplotype B is more variable, as it contains from one to five activating KIR receptors. Currently, more than 40 haplotypes B have been described. The function of NK cells regulated by signals from KIR receptors is driven by HLA as KIR ligands are HLA class I, HLA-C and HLA-B. HLA molecules were grouped into four major categories based on the amino acid sequence determining the KIR-binding epitope. All expressed HLA-C alleles are placed in C1 or C2 group and most HLA-B alleles can be classified as either Bw4 or Bw6. The KIR2DL2 and KIR2DL3 inhibitory receptors and the KIR2DS2 activating receptor bind HLA-C1 (HLA-Cw*02, Cw*04, Cw*05, and Cw*06), while the KIR2DL1 inhibitory receptor and the KIR2DS1 activating receptor bind HLA-C2 (HLA-Cw*01, Cw*03, Cw*07, and Cw*08). The KIR3DL1 and KIR3DS1 bind the HLA-Bw4 [34] (Figure 4). Upon ligand/receptor coupling, depending on the signal, NK cells will be activated or will not react to the target cells. Therefore, this information could be used to predict NK cell responses and possible treatment modifications. Of interest, a IPD-KIR ligand calculator and a donor KIR B-content group calculator have been developed using the KIR-Ligand mismatch strategy to predict the responses but these are only research tools not used in clinical transplantation [35].

## 4. KIR Typing Methods

Identification of KIR genes developed so far include flow based and a PCR based method (Table 1).

In the flow method, commercially available anti-KIR/CD158 family antibodies are used in studies of KIR expression on NK cells [36,37]. This method has one major advantage—it accurately shows the activation state of NK-cells at the protein level by expressing specific KIRs, and not just an assay at the gene level. 

Nevertheless, currently the most widely used methods for KIR gene typing are polymerase chain reaction sequence-specific oligonucleotide (PCR SSO) and polymerase chain reaction with sequence-specific primers (PCR SSP). These strategies provide information in low or intermediate resolution, genotyping of the KIR gene system by means of determining the presence or absence of genes. 

The PCR SSO typing procedure is based on the hybridization of a labeled, single-stranded PCR product to specific oligonucleotide probes. A key step in this method is the PCR amplification of DNA with specific biotin-labeled primers. The resulting products undergo a hybridization reaction with specific oligonucleotide probes. This reaction produces a color signal that is detected by the analyzer Luminex. Analysis of all signals from the responding probes in special program allows to obtain the result of KIR genotyping on intermediate resolution (Figure 5) [38,39].

The PCR SSP method is based on a PCR reaction with specific primers. Amplified DNA fragments are separated according to size, e.g., by agarose gel electrophoresis, then visualized with the aid of intercalating DNA dyes and then exposure to UV light. The interpretation of the PCR-SSP results is based on the presence or absence of the specific product and compared with the iterative tables and analyzed in the supplied software. The result obtained is at the low resolution level [40,41,42].

Real time PCR is also a very useful method. The use of SYBR labeling or specific molecular probes allows us to skip the electrophoresis stage, which makes this method easier to apply in the laboratory [43,44].

Recently, in addition to the methods used to evaluate KIR receptors, the *next generation sequencing* (NGS) method has been added, thanks to which we can obtain a high-resolution result [45].

In summary, KIR high-throughput genotyping methods are now widely available, relatively inexpensive and may increase the pool of fully characterized donors and recipients in the transplant procedures.

## 5. NK, KIR and Hematologic Diseases

Treatment of patients with certain high-risk hematological diseases is only possible with stem cell/bone marrow transplantation. Selecting the best donor is a complex process that depends on many factors. The key criterion in the cell donor selection procedure for a hematological patient is histocompatibility in HLA antigens between recipient and donor. According to the algorithm adopted by the EBMT 2019 (https://www.ebmt.org, accessed on 3 July 2021), the first choice is a fully compatible family donor. The probability of having a fully histocompatibility donor is 25% [46]. Especially for patients with acute myeloid leukemia (AML), HLA-matched sibling donors remain the best option due to rapid hematopoietic and immunological reconstruction, and a lower incidence of infections and acute graft-versus-host disease (aGVHD) [47]. If the patient does not have a fully compatible family donor or another reason (transplant as a rescue, no siblings etc.), other options may be considered. (1) Transplantation from a haploidentical donor- a donor who has a common HLA haplotype confirmed by high-resolution methods (parents, child, siblings). Cross-matching and anti-HLA antibodies are also required [48]. (2) Selection of unrelated donor from world bone marrow donor registry *(*BMDW*)*. Full compliance within all HLA-A*, B*, C*, DRB1* and DQB1* on a high resolution are preferred first. The following other criteria should be used in choosing the best donor: sex (if possible, choose a male, if a woman, then those who have not yet given birth), age (a younger donor is better), CMV status, blood group compatibility and optionally permissive mismatching in HLA-DP. Donors for whom the patient has donor specific antibodies (DSA) are avoided [49]. Different procedures are used depending on the individual situation of the patient and the experience of the transplant center. (3) Umbilical cord blood (UCB).

Given the compatibility of the HLA antigens, a fully compatible donor is the best choice, but unfortunately some patients do not have such a donor and then a non-fully compatible donor is selected, the so-called alternative donor. This is especially important with regard to the mismatch at the C locus, which is one of the ligands for KIR receptors. This mismatch may affect NK cell specific activity after transplantation [50]. Some high-risk diseases are sensitive to NK cell activity. This is the post-transplant population that rebuilds first, even before T cells, and it may be primarily responsible for graft loss, GvHD response and post-transplant leukemia control. They have numerous effector functions, such as the secretion of the cytokines or cytotoxic activity, which play a significant role in cancer surveillance. This NK cell activity is one of the pillars of the GvL effect [51]. This reaction relates to the ability of the donor’s immune cells to eliminate host leukemic cells after *hematopoietic cell transplantation* (HCT) [52]. The effect is explained by the activation of the “missing self” hypothesis. Triggering of the cytotoxic activity of NK cells resulting from the HLA class I mismatch between the recipient and the donor of stem cells allows donor cells to destroy remaining leukemic cells and improve transplant treatment results. The donor cells have inhibitory KIR receptors, and no ligands for these receptors are present on the recipient cells vs. tumor cells [53]. NK cells are activated, which results in cytotoxic activity against leukemic cells and cell lysis (Figure 6). This anti-tumor activity strongly depends on the HLA matching. Recently, a group of scientists has presented on an inhibitory KIR-HLA calculator they developed in order to select the best donor. The possibilities of iKIR/HLA were presented in three groups: favorable, adverse and unchanged. Their results showed that the number of iKIR-HLA pairs after HSCT has an influence on risk of recurrence and the overall survival [54].

Currently, research on the importance of the KIR/HLA relationship in the field of transplantation is still in progress and opinions in this area are disparate and widely discussed. A pioneering report published in 2002 by Velardi and colleagues proved in a mouse model and in clinical trials the importance of a KIR ligand mismatch between donor and recipient and its impact on the protection against relapses. An adverse reaction of the transplant recipient is GvHD caused by antigenically introduced foreign lymphocytes. The donor T lymphocytes infiltrate the recipient tissues and organs, leading to their destruction. The infiltrates are most often in the skin, the digestive tract and mucous membranes. Reactions can be very severe and the risk of death for the patient is sometimes high. GvHD can be reduced by depleting T cells, but then the risk of transplant rejection increases. The authors showed in their studies that the KIR/HLA mismatch of the donor-recipient after the removal of T cells from the graft material in patients with (*acute myeloid leukemia)* AML after transplantation contributed to a significantly lower rate of relapses. There were no such spectacular effects in patients with *acute lymphoblastic leukemia* (ALL). In addition, they presented the results of an experiment carried out in a mouse model. They infused alloreactive NK cells using a protocol with reduced conditioning intensity and obtained satisfactory transplant results—reduced GvHD and relapse. In their view, even alloreactive transformed NK cells are associated with total donor chimerism and leukemia eradication [55]. These studies inspired numerous subsequent studies around the world and opened the doors to novel diagnostic and therapeutic possibilities. Numerous teams confirmed the findings but there are also reports which did not find such associations [56]. For example, Zhao et al., prospectively investigated 2 groups of patients: *n* = 114 (AML, ALL, *chronic myeloid leukemia*-CML, *myelodysplastic syndrome*-MDS) and *n* = 276 (AML, MDS), who received haploidentical *stem cell transplantation* (haplo-SCT). Researchers investigated the relationship between the presence of ligands in the recipient and donor for the donor KIR receptors and NK activity. Total NK cell count and anti-K562 (Lymphoblast human cell line) response were tested by examining CD107 expression levels and IFN-γ expression, among others. The final conclusion of this study was that the presence of ligands for donor KIR receptors in recipients resulted in lower levels of relapses [57]. Recently, Weisdorf et al., published the results of the relationship between the haplotype of KIR genes in the donor and the results of transplantation in a cohort of 243 patients. Haplotype A contained a set of genes encoding mostly inhibitory receptors, except one. On the other hand, haplotype B was characterized by a set of mainly activating genes. The presence of haplotype B in donors of hematopoietic cells resulted in significant protection against disease recurrence and improved disease-free survival after reduced intensity conditioning, but only in recipients with C1 antigens [58]. This study was a continuation of many earlier observations from previous years, where they showed statistically better transplant results and significant differences in patient survival on a large number of patients, in pairs where the donor had the B haplotype [59,60]. The mismatch between KIR and HLA was also investigated by Wanquet et al., who studied 144 patients with hematologic malignancies treated with T cell—replete (without removing T cells) haploidentical stem cell transplantation (Haplo-SCT) with the use of post-transplant cyclophosphamide (PT-Cy). They observed that, depending on the disease state over time after Haplo-SCT, the clinical effect of the KIR ligand mismatch (KIR-Lmm) may be different. Among patients with active disease, with no complete remission (CR), they found a strong reduction in the number of relapses and better results in remission status, however, without the influence of GvHD and non-recurrence mortality. In contrast to the no CR group, they did not observe any difference in the patients in the CR group. This may be related to the earlier reconstitution of NK cells than T cells. It seems that NK cells are involved in GvL in the absence of a ligand for KIR in the recipient. Such information could increase disease control and individualize the treatment process. Moreover, the presence of KIR-Lmm was not associated with a significant increase in the incidence of GvHD, because NK cells can induce the GvL effect while avoiding GvHD [61]. The latest results of Varneris’s research team are of particular interest. Taking into account that so far many cases and relationships between alloreactivity of cells and NK and treatment of hematological neoplasms in adult patients have been described, the authors have focused on the pediatric group that received treatment in the form of allogeneic transplant from an unrelated donor. They retrospectively examined a total of 716 patients, of which *n* = 372 with ALL and *n* = 344 with AML. Scientists performed KIR typing, KIR ligand mismatch, KIR2DS1 and Cen B/telomeric A mismatch in the recipient donor pairs. The aim was to find out if any variables relevant to KIR were related to relapses in children with AML or ALL. Unfortunately, they found that none of the mismatches were significantly associated with relapse or disease-free survival, considering the entire patient cohort (ALL and AML) and separately AML or ALL. There was also no relationship between the KIR variables and the results in the group of transplantation with T cell depletion. The authors, taking into account these findings, did not recommend in the donor selection a consideration of the KIR/HLA matching in children [55]. Similar conclusions were reached by scientists from Japan, who also, in a study of 137 children with ALL-B, did not find a significant difference in the outcome of the transplant and KIR matching [62].

Haploidentical transplantation was a revolution. Currently, the results of these transplants are continuously improving but they are still not fully satisfactory. There are still no strict criteria for selecting a donor if there are more candidates. In such cases, excluding the donor against whom the patient has DSA antibodies, the donor-mother and preference for the male gender are the most chosen recruitment factors. Furthermore, the issues of developing special desensitization protocols for viral or bacterial immunized patients, so that the level of DSA antibodies are safe for the entire transplant procedure, are also increasingly discussed. In the study by Solomon et al., 208 transplant selections were analyzed. The donor variables analyzed were age, gender, relationship, CMV status, ABO compliance, HLA discrepancy and KIR mismatch. The overall 3-year survival rate was analyzed depending on the data characteristics of the donors. Incompatibilities in the KIR-like receptors resulted in overall *survival* (OS) (65% versus 51%). The presence of the KIR B/x haplotype with the activating KIR2DS2 receptor was 69% vs. 54% vs. 46% vs. to KIR A/A and KIR B/x, without 2DS2. The presented data show that the mismatch in KIR genes contributed to the reduction of the number of progressions and relapses. Additionally, the presence of the KIR B/x and KIR2DS2 haplotype improved the results on non-relapse mortality. The result of these analyses is the evidence-based system for assessing donor risk factors proposed by the authors, in the form of a point selection algorithm for choosing the best haploidentical donor individually for each patient. This comprehensive analysis clearly demonstrates the importance of genotyping of KIR genes in addition to other clinical parameters, and should be routinely performed as part of the best haploidentical donor selection procedure to optimize survival and reduce the risk of relapse [48]. Currently, it is the only proposed point algorithm that takes into account the dependence of KIR/HLA in haploidentical transplantation procedures. It seems that the presented algorithm is perfectly applicable. The potential benefits and the currently low cost of typing KIR receptors prompts the use of this algorithm in routine work. In the case of related donors, the analysis of KIR and HLA receptors may also have a beneficial effect on the results of stem cells transplantation. In 2020, an interesting report presenting the expression levels of specific KIR genes in 252 donor-recipient pairs in stem cell transplantation was published. The method used to evaluate expression was RT-qPCR. The KIR genes subjected to the postponed comparison were selected so that they appeared only in the donor before transplantation. The obtained results indicate that the expression levels of the KIR2DL2 gene were recorded at 3 months after transplantation and the measured levels were higher than in the donors. It was similar with KIR2DL1 and KIR3DL1 where the expression level was significantly higher at 2 and 3 months after transplantation compared to the levels determined in the donor. This study shows how the level of expression of KIR genes and, subsequently the production of functional proteins after stem cell transplantation, can be of great importance. This knowledge could contribute to the ongoing monitoring and prediction of post-transplant effects [63]. Recently, Solloch et al. published a study pointing at the polymorphisms of the KIR gene family. Currently, in the field of HLA, the frequency of haplotype data has been developed for concurrent populations, which is helpful in the search for the best donor. Considering the potential benefits of understanding the KIR/MHC matching, the researchers focus on the prevalence of the population-specific KIR gene haplotypes. For example, in Germany 458 families were screened for HLA (High resolution), KIR, blood group AB0, CCR5 co-receptor and major histocompatibility complex class I chain–related molecule A/B (MIC A/B). On this basis, a linkage disequilibrium table was developed with regard to 2 loci and also a larger number, and the frequency of the 20 most popular haplotypes of the KIR allele groups was estimated. Such a study seems to be useful in understanding the structure of KIR genes which may result in a better results at the stage of the donor selection procedure [64].

## 6. NK, KIR and Solid Organ Transplantation

The best option for a patient with end-stage disease is an organ transplantation. Due to the fact that there are few transplants from living donors (kidney), and a shortage of organs from deceased donors, the donor–recipient pair should be selected as carefully as possible so that the transplanted organ survives as long as possible. It is the incompatibility between recipient and donor regarding HLA antigens that leads to the stimulation of the recipient’s immune system and rejection of the transplanted organ. The advances in immunology diagnostic, such as HLA matching, PRA, detecting of alloantibodies, especially DSA and complement binding (ab anti-HLA C1q), cross match -CM-CDC, DSA monitoring, IgM detection and novel immunosuppression strategies have reduced the incidence of rejection and improved survival results in solid organ transplantation patients. Although it does not totally prevent the progressive loss of function of the graft, it allows for a very long functioning of the transplanted organ. Nevertheless, there are still unanswered challenges in this area. The antibody-mediated rejection (ABMR) is the largest reason for a late graft loss. There are three types within the antibody-mediated mechanism: hyperacute, acute and chronic. Hyperacute rejection is mediated by recipient’s preformed DSA donor specific antibodies. It is observed immediately after perfusion of transplanted organs by recipient’s blood. Alloantibodies can bind antigens on the vascular endothelium and cause tissue damage. Prospective complement-dependent cytotoxicity—cross match (CDC-CM) minimizes this type of rejection.

Novel reports (2019) diminished the doctrine that microvascular inflammation during rejection (MVI) is caused exclusively by the antibody-mediated pathway [65]. For this purpose, kidney allograft biopsies were performed in 938 patients, 129 of whom had MVI. Subsequently, all MVI patients were assessed for their DSA levels. About half of these patients had no DSA. According to the authors, NK cells may also be responsible for MVI. Incompatibility between MHC and KIR may induce alloreactivity of NK cells according to the “missing self” hypothesis and lead to damage in the transplanted organ. Moreover, it was suggested that this “missing” NK activation was the mechanistic target of rapamycin (mTORC1) dependence, referring to previous research and experiments in a mouse model. These results lead to the conclusion that the presence of NK cells in chronic rejection could be related either to humoral rejection or, independent of the antibodies and mTOR inhibition, might be applied to protect the evolution of this type of chronic vascular rejection [65]. In many countries, the standard-of-diagnostic in recipient–donor matching, HLA (A*,B*,DRB1*) typing does not take into account C* alleles, which are the main ligands for KIRs. The introduction of the KIR designation in the transplant diagnostic would be a novelty. La Manna et.al studied the effect of KIR and HLA immunogenetic systems on the long-term outcome of kidney transplantation [66]. They genotyped 126 patients after kidney transplantation within HLA-A,B,C,DRB1 and KIR genes. Statistically higher serum creatinine levels were observed in the group of patients lacking KIR2DL1. An interesting relationship has also been shown for KIR2DS3. The recipients with KIR2DS3 gene revealed a tendency towards lower creatinine levels and a higher estimated glomerular filtration rate (eGFR) compared to patients without the KIR2DS3 gene one year after transplantation. These differences were still statistically significant four years after transplantation. The KIR2DS3/ligand analysis detailed that recipients with KIR2DS3, who received a kidney with the HLA-C1+ ligand had higher serum creatinine levels and lower eGFR factor 1 year after transplantation. Interestingly, the authors also showed that kidney transplantation to a patient who has the KIR3DL2 gene from a donor who does not have HLA-A3/HLA-A11 is a protective factor influencing 5-year survival [66]. In another work, Littera et al., presented the link between chronic rejection and the correlation of KIR/MHC. The analysis on 174 patients revealed that a risk of chronic kidney rejection was higher in recipient–donor pairs other than the combinations rKIR2DL1/dHLA-C2 or rKIR3DL1/dHLA-Bw4 (r: recipient, d: donor). The absence of these two immunogenetic profiles led to low levels of NK cell inhibition, that is, increased cytotoxic activity of NK cells against the transplanted organ [67]. Therefore, patients with this high-risk profile could benefit from immunosuppression aimed at reducing NK cell alloreactivity. Moreover, an interesting finding also concerned the fact that transplanted patients from C1 homozygous donors had more chronic rejection incidents than patients with an organ with a homozygous C2 antigen or a heterozygous C2 donor, or the Bw4 ligand group.

Regarding lung transplantation, the major reason of this significant survival reduction is chronic lung allograft dysfunction (CLAD), which develops in about 50% of recipients within 5 years after lung transplantation [68]. The mechanisms underlying CLAD are not well understood. Recent publications focused on NK cells as modulators of transplant success in both acute and chronic graft loss may increase our knowledge. Greenland et al., found that bronchoalveolar lavage (BAL) obtained from patients with acute cellular rejection is enriched with NK cells. This enhanced infiltration with NK cells has been confirmed in allograft transbronchial biopsy specimens, notably in lung allografts undergoing rejection [69]. Among the others, NK cell function is dependent on the total charge of membrane inhibitory and activating receptors. This phenomenon has been suggested by Kwakkel-van Erp et al., who found that the risk of CLAD was higher in KIR A haplotype, which contains more inhibitory receptors than activatory ones [70]. Among 48 recipients who received a lung transplant, the KIR haplotype A and KIR2DS5 were predictive of the development of chronic transplant rejection. Another meaningful association was found between recipient’s KIR3DL1/Bw4 and the donor’s Bw6 alleles. This was associated with host-versus-graft effect but also with the decreased risk of CLAD, early bronchitis and death [71]. The authors showed greater CLAD-free survival in recipients with Bw4 who received lung allografts from donors with Bw6 as compared with others. This results suggested that recipient’s KIR3DL1 NK cells may be responsible for the killing of donor HLA-Bw6 *antigen presenting cell*s (APC) prior to an alloantigen-specific T cell response.

In contrast to lung transplantation, chronic liver transplant rejection occurs in about 3–4% of cases [72]. Reports on KIR/HLA class I interactions in liver transplantation are extremely rare. No significant associations between KIR genotype of patients and risk of AR were detected. Fosby et al. tested 143 donor-recipient pairs within 6 months after transplantation and the results did not indicate a mismatch in KIR/HLA genes as the cause of AR [73]. In contrast, Hyeyoung et al., demonstrated in a cohort of 182 liver recipients from living donors higher levels of AR in liver transplant patients, who received organs from C2 HLA donors, when compared to HLA C1 donors. They also showed a positive relationship between AR and the number of C2 group HLA alleles. Nevertheless, KIR genotypes and the number of KIR activating/inhibiting KIRs were not related to the allograft outcome and did not have a statistically significant effect on graft survival [74]. Legaz et al. reported that KIR2DL3 and KIR2DS1 in recipients correlated with the increased incidence of AR in the presence of donor C2 ligands. They considered the iterative KIR/HLA study as a target of further research as clinically relevant for protection against acute rejection and personalized treatment [75]. 

## 7. KIR/HLA Interaction and Association with Viral Infections after Transplantation

The ‘missing-self’ hypothesis implies that this mechanism of NK activity is of special interest in viral infections as viral antigens are often presented via HLA class I receptors. Moreover, viruses often use decreased expression of HLA receptors as a way to hide from T cytotoxic response. Transplantology is of special interest here as post-transplant immunosuppression results in a high-risk of viral infections. Cytomegalovirus (CMV) infection is the most common severe viral complication after solid organ transplantation (SOT). It remains a major source of morbidity and mortality in solid organ transplant recipients. Currently, only the donor and recipient serological status is used to predict the risk of infection after transplantation, while the cytotoxic potential of immune system is left unassessed. At the same time, NK cells are very much involved in the anti-viral response. NK antiviral activation is modulated by KIRs, so precisely the KIR receptors and HLA genotype could be associated with the risk of CMV disease. Deborska-Materkowska et.al presented the association between the lack of activating KIR2DS2 and lower CMV infection rate after transplantation [76]. The association between KIR2DS2, KIR2DS4 and CMV infection rate was described in hematology during stem cell transplant treatment [77]. Another significant finding was a strong association of KIR2DL3 and KIR2DS2 alleles with the time to CMV viremia [78]. In the same study CMV-negative recipients who carried KIR2DS2 and KIR2DL3 receptors, whose donors were CMV-positive, were protected against CMV viremia. This protective association was observed only if the donors did not express any HLA-C2 group HLA alleles. The significance of KIRs has also been presented by Behrendt et al. [79]. Another report showed that recipients who carried KIR2DL3 and KIR2DL2 were more often developing active cytomegalovirus infections [76]. According to Hadaya, et al., B haplotype in the recipient is linked to the protection from cytomegalovirus replication after solid organ transplantation [80]. There are studies which report that patients carrying the combination of the activating KIR3DS1 receptor and its ligand HLA-Bw4 have the advantage of delayed progression to acquired immunodeficiency syndrome, whereas the presence of KIR3DS1 in the absence of Bw4 is associated with a rapid progression. These data also suggest that virus elimination is facilitated upon binding of an activating KIR to its ligand [70]. Altogether, this highlights the importance of the balance between the activating and inhibitory signals in the activity of NK cells. Several studies have suggested that KIR genes may be also associated with the risk for other viral infections, such as hepatitis C, herpes simplex, BK polyomavirus (BKPyV), Epstein–Barr virus (EBV) and varicella zoster virus (VZV) [81].

## 8. KIR/HLA Interaction in Pregnancy Complication

Habitual miscarriages remain a problem for many couples. It is estimated that live births account for only 20–25% of fertilizations [82]. These occur more often due to women deciding to become mothers at a later age, complications of natural pregnancy or after in vitro fertilization. At least some of these problems seem to be a disorder of immune regulation. Uterine NK infiltrate the uterine mucosa and persist during normal pregnancy until delivery. It shows the importance of these cells not only for the implantation but also for the maintenance of pregnancy [83]. Uterine NK cells are close to the trophoblast cells and are therefore a determinant of maternal acceptance of the fetus. Uterine NK cells reduce cytotoxicity, which promotes the vascular formation of the arteries and enables trophoblast invasion by secretion of various cytokines, such as macrophage colony stimulating factor (M-CSF) and granulocyte-macrophage colony-stimulating factor (GM-CSF). They are necessary for the creation of an environment conducive to embryo implantation. Moreover, it has been shown that the secretion of growth stimulating factors from NK cells is essential for fetal growth [84]. Apart from the cytokine secretion properties, the properties of cytotoxic alloreactivity are also important. There is an hypothesis that maternal NK cells that carry a cytotoxic load are present in the uterus, where they recognize paternal HLA-C antigens on the surface of the fetus and trigger fetal damage. The balance between the regulation and activation mechanisms is crucial for the proper development of pregnancy. Many studies are focused on the role of NK cells in the implantation failure. Particular attention is focused on the KIR/HLA matching but the results are inconclusive. There are reports on the correlation between KIR AA haplotype, KIR BB and implantation failure/maintenance of pregnancy [85]. A recent study compared KIR haplotypes and the HLA-C genotype in the transferred embryo and its impact on pregnancy success. There have been 668 mothers with a single embryo transferred examined. The study revealed that KIR2DS1, KIR3DS1 and KIR2DS5 haplotypes were more frequent in spontaneous abortions and patients with the KIR A haplotype showed a lower risk of pregnancy loss compared to carriers of the KIR B haplotype. However, among the group of patients with the KIR A haplotype, the risk of pregnancy loss was significantly influenced by the presence/absence of the C1 allele in the embryo. The combinations (KIR A/homozygous C2 and KIR B/homozygous C1) led to up to 51% greater risk of loss compared to other combinations [86]. Similarly, the study by Soheil Akbari showed a significant association of maternal KIR2DS1 in combination with paternal HLA-C2 as a risk factor of spontaneous abortion [87]. Alomar et al. recently performed a study in which KIR/HLA associations were studied in recurrent spontaneous abortions (RSA). Sixty-five healthy women with a history of RSA (three or more spontaneous abortions) and 65 healthy controls (with at least two healthy children) were analyzed. The frequencies of KIR2DS2 and KIR2DL5A were significantly lower among women with RSA compared to the group of healthy women. No association with maternal HLA-C genotypes was observed. The analysis of the KIR/HLA-C combination showed the protective effect of KIR2DS2 with the HLA-C1 ligand. There are also reports that the KIR genes of haplotype B may have an impact on pregnancy success [88]. The possible participation in the initiation of pregnancy has been linked with the presence of the KIR2DL4 receptor. However, it is not conclusive as positive and no-effect studies can be found in the literature [89,90]. A recent meta-analysis of data from articles from the databases like Web of Science, PubMed, Scopus, Google Scholar, etc. was published by Iranian scientists. They concluded that KIR3DL1 was significantly linked with the protection from RSA. On the other hand, KIR2DS2 and KIR2DS3 alleles presented significant risk factors for RSA. There was no uniform conclusion for KIR2DS1 [91].

## 9. KIR/HLA Interaction in Solid Cancer

In order to escape immune surveillance, cancer cells have developed multiple escape mechanisms. One of them is to reduce the expression of MHC molecules on the surface, which hide the cancer cells from the detection by cytolytic T lymphocytes. On the other hand, this fact is used by NK cells. NK cells mechanism of action is based, inter alia, on recognizing and attacking cells with weak or low HLA expression. The decreased expression of MHC by tumor cells results in a greater cytotoxic sensitivity of NK cells. The control of NK cell reactivity is a function of the signals coming from the interaction of NK cell receptors and MHC ligands. Hence, the knowledge of NK and MHC cell receptors seems to be of interest and may constitute a therapeutic point in the treatment of solid neoplasms. KIR receptors are particularly involved in the regulation of NK cells. When the net result of the interaction between KIR and their MHC ligands is negative, NK cells are less sensitive, which is unfavorable in the fight against cancer. If there is an NK activation prevalent, NK cells are activated and eliminate cancer cells. Recently, numerous studies have shown a link between KIR receptors and their ligands, and the protection or susceptibility of solid tumors [92]. For example, a paper examining the relationship between KIR and HLA among patients with non-small-cell cancer NSCLC was only recently published. In total, 229 patients from the Chinese population were examined [93]; no differences were found between the frequency of KIR or the KIR/HLA combinations among the controls, while there was a more frequent occurrence of HLA-C*08:01 among patients. In another study, a relationship between the increased frequency of the KIR2DL1-C2 genotype and the occurrence of NSCLC was found [94]. It also turns out that the expression of some KIR receptors correlates with poor prognosis and clinical characteristics [95]. Other tumors may also depend on KIR/HLA matching. For example, a higher incidence of KIR3DL1 and the risk of the disease was reported in basal cell carcinoma [96]. The KIR genes were analyzed in gastric cancer and it was found that 2DS1, 2DS3, 2DS5, 3DS1 and 2DL5 are associated with overall survival, and the presence of haplotype B was also found as a risk factor [96]. In another cancer—neuroblastoma, the presence of individual KIR genes, KIR2DL2 and KIR2DS2, regardless of the HLA-C genotype, may predict treatment outcomes [97]. In the case of cervical neoplasms the alleles of the HLA-C1 group may be important in protection against HPV16-associated cervical tumors, mainly through the regulation of the KIR/HLA interaction [98]. There are also studies that show no correlation whatsoever, for example, in prostate cancer [99]. The topic on the interaction of KIR/MHC receptors is not exhausted and further studies are needed. The interactions may be the starting point for cancer therapy [100] and may be a criterion for selecting an appropriate treatment.

## 10. Therapy

Using the conclusions and the described models of NK cell alloreactivity flowing from numerous centers around the world, numerous concepts of adoptive NK cell therapies are currently being developed. Miller’s team research was one of the first attempts with partial success performed in 2005. The team had shown in previous studies that NK cell infusion post HSCT was safe but failed to ignite the full GvL effect. In the studied group, there was no matching of the inhibitory receptor, which could explain the lack of improvement of the treatment effects [101]. In another study, the same group of scientists tested haploidentical NK cell infusions from related donors. Two groups of patients were studied in a non-transplantation setting: the first with low doses of cyclophosphamide and methylprednisolone, and the second with fludarabine. Patients with acute AML and poor prognosis received a high dose of cyclophosphamide and fludarabine. After NK cell infusions, all patients received IL-2. It has been observed that patients treated with a lower intensity did not expand donor NK cells in vivo. Such an expansion of NK was noted in the group under intensive treatment. Endogenous IL-15 increased in vivo expansion of donor NK cells and CR was noted in five out of 19 patients with high-risk AML. This work proved that the administration of haploidentical NK cells can influence the treatment and can be used as an additional therapeutic tool [101]. Another aspect of adoptive therapies is the use of cytokines when culturing NK cells ex vivo. One of the most widely used and effective cytokines is IL-15 as it has the greatest influence on the development of NK cells. Shah et al. demonstrated studies which show that IL-15 enhances NK cell activation and proliferation, and contributes to anti-tumor activity [102]. In the study of the Pittsburg group, it turned out that the alloreactivity of NK cells against autologous AML leukemic blasts was significantly higher in the case of pre-activation of NK cells with IL-15 [103]. While the studies using NK cells have given hope for AML, a group of scientists has focused on studying patients with ALL. For this purpose, Roser’s leukemia, which is the model for-T-ALL, was used. It was found that the originally attenuated blast cell activity was increased by pre-activation of NK cells with IL-12, IL-15 and IL-18. This resulted in an increase in the expression of activation receptors on the surface of NK cells and a delayed development of neoplastic cells [104]. Another way to use the cytotoxicity of NK cells in the treatment of cancer is through the activation of antibody-dependent cell cytotoxicity (ADCC). The CD16 molecule on the surface of NK cells binds FcIgG fragments of monoclonal antibodies, such as Rituximab, on the surface of D target cells, enhancing the effect of killing cancer cells. Going even further, based on the success of chimeric antigen receptor (CAR T) lymphocytes, NK cells, by virtue of their cytotoxic activities, are also a good target for CAR design and development of CAR-NK therapies. In ongoing clinical trials, anti-CD7 CAR-expressing NK92 cells are tested in patients with acute myeloblastic leukemia and T-cell tumors (NCT04004637) (NCT02742727), and B-cell lymphoma or leukemia. (NCT03056339); the results of which are yet to be available. In the last year, the results of studies on the effects of chimeric NK cells from the NK92 line with missing inhibitors of KIR receptors on neoplastic cells were published. The conducted experiments showed a strong activity against lymphoma cells [105]. Recent developments in the therapy are aimed at optimizing NK cell therapy in the fight against blood cancers by blocking KIR inhibitory receptors. Such actions can lead to maximal anti-tumor effects by inhibiting the flow of inhibitory signals. This is achieved with biological treatment using anti-KIR monoclonal antibodies, such as Lirilumab. Lirilumab is a blocking antibody directed against the KIR2DL1, KIR2DL2 and KIR2DL3 NK receptors [106]. There are many studies evaluating this drug administered alone or in combination with other immunomodulatory anticancer drugs. As a single, in the phase 1 trial, it showed a satisfactory level of safety, good clinical tolerance of patients and a permanent blockade of KIR in patients with solid tumors and hematologic malignancies [107]. Lirilumab was also tested in combination with azacitidine. This drug combination shows clinical activity in MDS patients. However, the authors indicate the need for further research (the study is in phase two) [108]. In combination with rituximab in patients with CLL in the refractory/relapsed after prior therapy cohort, the mortality rate was lower than the untreated with high-risk molecular features group 0% vs. 16% (NCT02481297). It is interesting to study the efficacy of anti-KIR monoclonal antibodies in maintenance therapy in acute myeloid leukemia (EFFIKIR) where the drug was administered to groups at different doses 0.1 mg/kg and 1 mg/kg (NCT01687387). The higher mortality rate was in the group that received 1 mg/kg—62.75% vs. 52%. Next, phase I studies have shown that IPH2101 (anti-KIR2D) is safe and tolerable in patients with advanced MM and it could be a promising therapy for these patients. These results lend support to further development of anti-KIR mABs as a novel therapy (NCT00552396, https://clinicaltrials.gov/ct2/show/NCT00552396, accessed on 3 July 2021) [109]. Similarly like in (NCT00999830). IPH4102 (anti-KIR3DL2 mAb) has also been evaluated and is in phase II clinical trials for the treatment of cutaneous T-cell lymphoma patients. IPH4102 is safe and shows encouraging clinical activity in patients with relapsed or refractory cutaneous, especially those with Sézary syndrome. If confirmed in future trials, IPH4102 could become a novel treatment option for these patients (NCT02593045) [110]. The combination of anti-KIR monoclonal antibodies with anti-PD-1/PD-L1 monoclonal antibodies is also promising, which could become a new therapeutic tool in blocking tumor immune escape in NSCLC [111]. The use of Lirilumab with anti-CTLA-4 antibody (Ipilimumab) in subjects with selected advanced tumor is assessed in the phase 1 studies. Blocking inhibitory signals may increase anti-tumor activity of NK cells or T-lymphocytes (NCT01750580). In ongoing clinical trials it is concluded that the antibodies used to block KIR receptors are safe and do not lead to an autoimmune reaction (NCT01714739). The use of anti-KIR monoclonal antibodies has been extensively tested to regulate NK cells. The use of bispecific (BiKE) and trispecific (TriKE) antibodies in augmenting antitumor activity seems to be extremely interesting [112,113].

Adoptive therapy using NK cells appears to be promising. However, the current high costs and organizational effort do not allow for daily use of such sophisticated cellular therapies. In addition, apart from reduced costs, appropriate treatment protocols complementing chemotherapy should also be developed.

## 11. Conclusions

Despite the PubMed database identifying over 5000 publications on the importance of KIR in the field of medicine since the 1990s and in consideration of the above reports, the area of research on KIR receptors remains insufficiently understood. The results obtained thus far are still inconclusive in many aspects. This is most likely due to the lack of multi-center studies and meta-analyses. Although the awareness on NK cell activity and the KIR/MHC relationship in the clinical management of the patient is rising, this information is still considered auxiliary, not primary. This information can be very useful in selecting the final donor or in order to obtain the best possible patient outcomes and benefits in the therapeutic process. This information can be an auxiliary parameter in the treatment of infertility, or ultimately it can be used in the prediction of antiviral reactions, which are the main cause of post-transplant mortality. The low cost of the testing, in view of the potential benefits, should encourage wider research on KIR receptors. The results will be certain to place this diagnostic procedure next to HLA antigens in the applicable standards.

## Figures and Tables

**Figure 1 cells-10-01777-f001:**
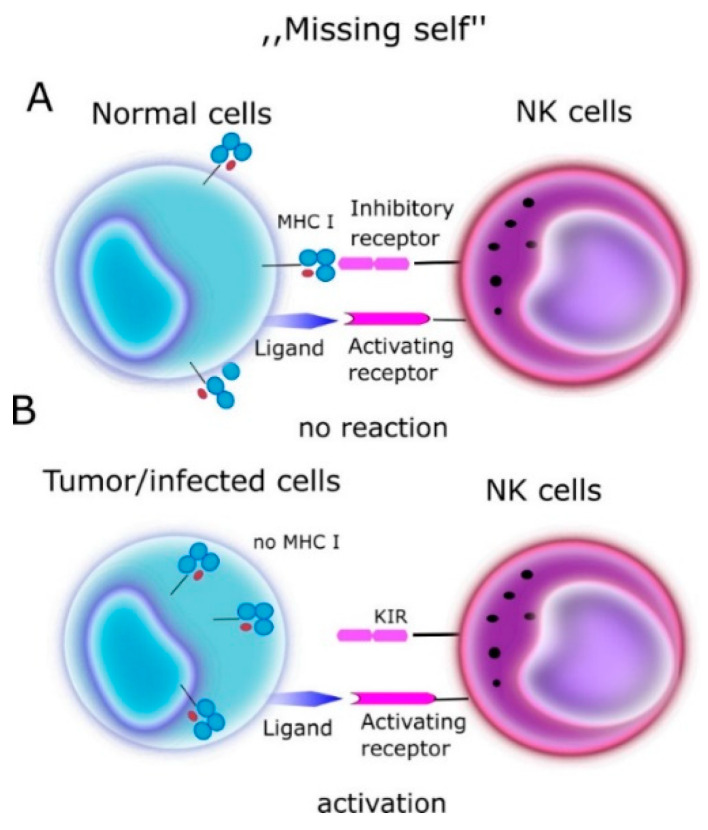
“Missing self” hypothesis according to NK cells kill cells with weak or no MHC-I expression. MHC molecules play a protective role against normal host cells, protecting them against lysis through NK cells. (**A**) On a normal cell there are MHC antigens (ligands for KIR)—no reaction. (**B**) On tumor/infected cells no MHC- NK cells activation.

**Figure 2 cells-10-01777-f002:**
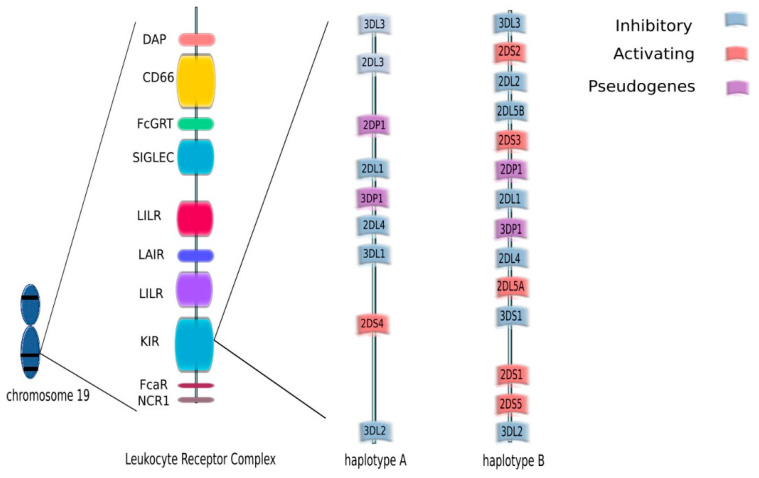
KIR genes are located on chromosome 19, in the leukocyte receptor complex (LRC). The KIR gene family consists of 15 gene loci (KIR2DL1, KIR2DL2/L3, KIR2DL4, KIR2DL5A, KIR2DL5B, KIR2DS1, KIR2DS2, KIR2DS3, KIR2DS4, KIR2DS5, KIR3DL1/S1, KIR3DL2, KIR3DL3) and two pseudogenes (KIR2DP1 and KIR3DP1.) KIR genotypes can be divided into two haplotypes: A and B depending on the composition of their genes.

**Figure 3 cells-10-01777-f003:**
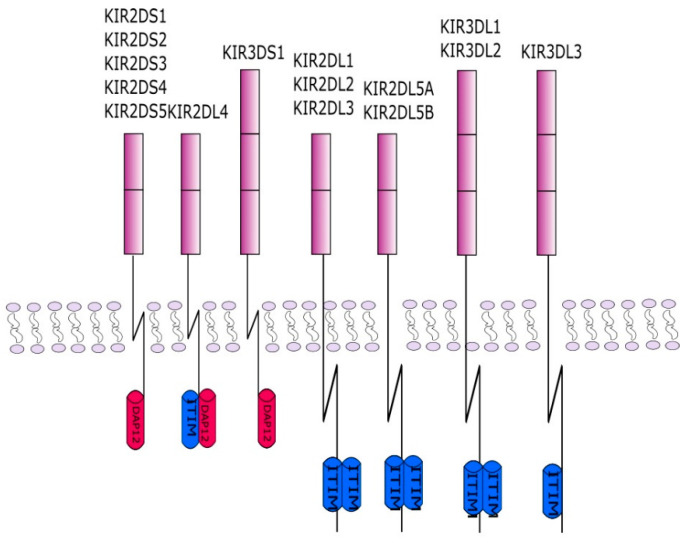
This figure shows the KIR protein structures. Each of the receptors consists of two or three protein domains and a long or short cytoplasmic chain. The chain in inhibitory KIRs is associated with an ITIM inhibiting sequence. The activating KIRs interact with the adapter DAP12 containing ITAM sequences. The interaction takes place through a charged amino acid in the KIR transmembrane domain.

**Figure 4 cells-10-01777-f004:**
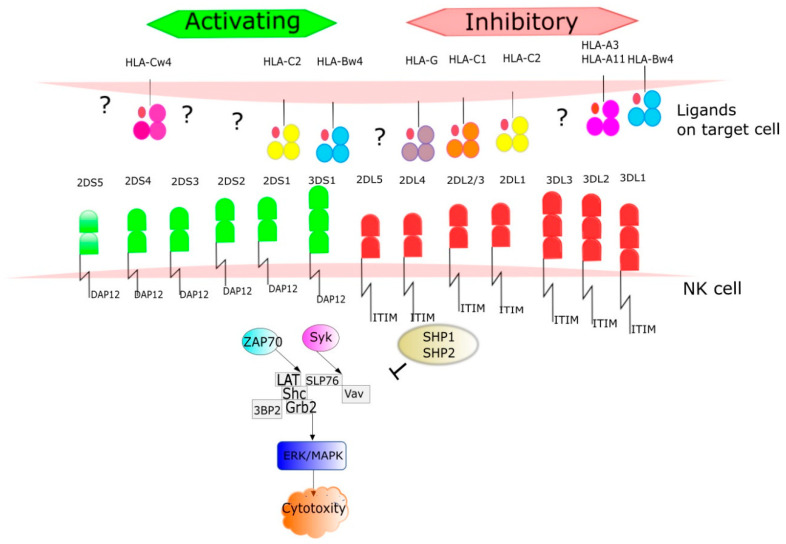
The ligands for KIR receptors are mainly HLA I antigens. HLA molecules were grouped into four major categories based on the amino acid sequence determining the KIR-binding epitope. All expressed HLA-C alleles are placed in C1 or C2 group and most HLA-B alleles can be classified as either Bw4 or Bw6. The number of protein sequence variants characterized to date for each KIR receptor is provided. The figure shows signaling pathways from the activated receptors via tyrosine kinase binding protein-DAP12 and inhibitory receptors to. Phosphorylation DAP12 events result in recruitment of SYK family kinases such as Syk (spleen tyrosine kinase) and ZAP70 (zeta-chain associated protein kinase 70 kDa). In this pathway take part via alia LAT (Linker for activating of T cells), Shc (Src homology 2 domain containing), GRB2 (Growth factor receptor-bound protein 2), 3BP2(c-Abl Src homology 3 domain-binding *protein*-2). Syk interacts with SLP76 (SLP-76 tyrosine phosphoprotein) and activates Vav proteins (guanine nucleotide exchange factors). ZAP70 and SYK pathways lead to ERK (extracellular signal-regulated kinases) and MAPK (mitogen-activated protein kinases) activation results in NK cells cytotoxity. Once the ligand is recognized by an inhibitory receptor, the pathway is blocked through dephosphorylation by tyrosine phosphatase protein (SHP-1) and 11 (SHP-2).

**Figure 5 cells-10-01777-f005:**
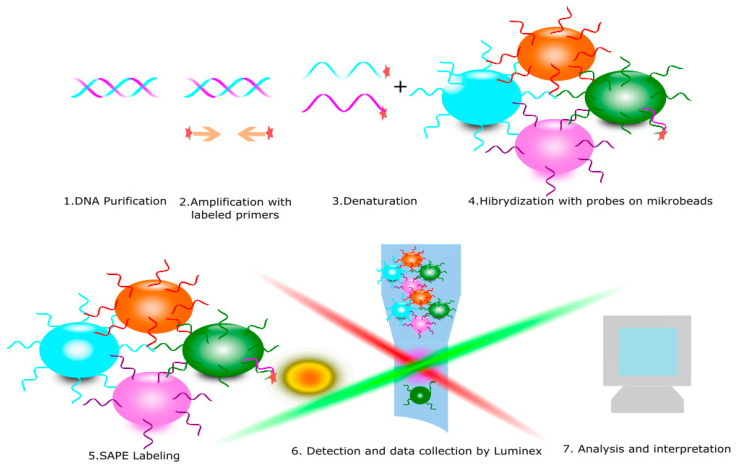
Polymerase chain reaction sequence-specific oligonucleotide (PCR SSO) is diagnostic method for KIR receptors typing. It consists of the following steps: 1. DNA purification, 2. Amplification with specific labeled primers, 3. Denaturation to obtain a single strand of DNA, 4. Hybridization with microbeads with specific probes and 5. Read and data collection.

**Figure 6 cells-10-01777-f006:**
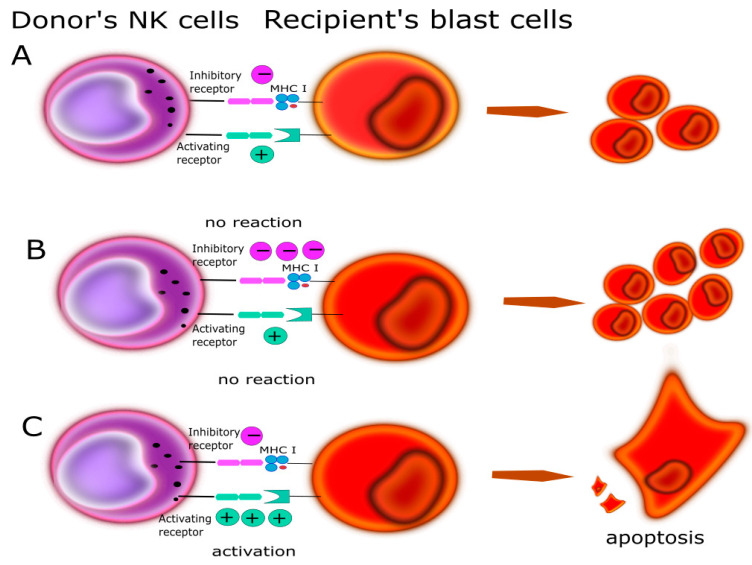
Relationship between KIR/HLA antigens could play a main role in GvL determined by the balance of activating and inhibitory input. (**A**) The recipient and donor are matched in a way that the final charge is balanced and does not induce cytotoxic activity in NK. (**B**) The recipient and donor are selected that does not induce NK cytotoxic activity; more inhibition signals. (**C**) The recipient and donor are selected that NK cytotoxic activity and leads to kill blast’s cells.

**Table 1 cells-10-01777-t001:** Methods for KIR typing.

Methods for KIR Typing	Based on KIR Expression on NK Cells	Based on DNA	Resolution Level	Quantitative	Qualitative	Time of Performing
Flow cytometry	+	−	−	+	+	<24 h
PCR SSO	−	+	intermediate	−	+	<24 h
PCR SSP	−	+	low	−	+	<24 h
Real time PCR	−	+	low	+	+	<24 h
NGS	−	+	high	−	+	>24 h

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
