# Peer review of "KIR Receptors as Key Regulators of NK Cells Activity in Health and Disease"

_cells, 2021, doi:10.3390/cells10071777_

Round 1
Reviewer 1 Report
The review article by Dębska-Zielkowska J. and collaborators presents the issue on KIR receptors and the regulation of NK cell function in a very detailed, broad and didactically way. The authors were particularly successful in addressing the different areas involved with this issue, as well as, the main methods to assess KIR-HLA relationship.
Author Response
Thank you very much for your opinion. We are very pleased.
Reviewer 2 Report
The presented review aims to describe the role of KIR molecular in health and disease. The topic is very relevant and complete information could be very useful. Unfortunately, this review is incomplete and requires major revision before consideration. Generally, there is much information (research data or facts) without proper citation, some paragraph contains very basic facts and are written in very basic language without scientific input. There are also some factual errors or misinterpretation which have to be definitely corrected. In particular, I found these weaknesses in the publication:
Line 49-50 There is a confusing interpretation. The role of NK cell is not to kill allogeneic tissue (or solid organs). They can take part in destruction, but it is not their role like antivirus or antitumor response.
Figure 1 is absolutely unnecessary and moreover it is completely unprofessional image showing gating strategy that doesn't really fit into the review.
Line 83-91 is with no reference, although it contains factual information.
Line 92-103 - the first and fifth points do not belong to the mechanisms of NK cell activity, but rather to characterization. I recommend excluding these points.
The last passage about NK cells is very confusing and the section jumps from function thought characterization, education, role in diseases memory and so on. It is really hard to read and not to be lost. The description has to have order, and the paragraph has to follow each other.
Line 222 – the newest database is from December 2020!
Chapter 4 – Typing methods – I really do not understand the inclusion of this section and the detailed description of methods on very low level of scientific language. The end of this section (last paragraph) does not follow up very much, and it is unnecessary.
Chapter 5 – sibling donors are not defined as the best donors. There are many criteria for the definition and selection of the best donor and this is not the one of them. The siblings can be haploidentical and therefore, the looking for HLA-matched donors can begin even in case of the existence of siblings. The selection of donors is complex process and is described in the review primitively.
Followed description of NK cells function are with no references and proper explanation what really happening after HSCT. The figure 6 is very similar to figure 2 and it is unnecessary to repeat the information (very basic and known information) in two sections .
Chapter six can be shortened, and the individual chapters may be common. It would be a good idea to shorten the introduction to reactions that affect KIR receptors and NK cells with a focus only on the relevant topic.
The last chapter about cellular therapy is focused mostly on NK cells and not fully focuses on KIRs. There are several preclinical and clinical studies using blocking of KIR molecules and they have to be described here.
Author Response
The presented review aims to describe the role of KIR molecular in health and disease. The topic is very relevant and complete information could be very useful. Unfortunately, this review is incomplete and requires major revision before consideration. Generally, there is much information (research data or facts) without proper citation, some paragraph contains very basic facts and are written in very basic language without scientific input. There are also some factual errors or misinterpretation which have to be definitely corrected. In particular, I found these weaknesses in the publication:
Line 49-50 There is a confusing interpretation. The role of NK cell is not to kill allogeneic tissue (or solid organs). They can take part in destruction, but it is not their role like antivirus or antitumor response.
As suggested, the sentence has been made more precise.
Figure 1 is absolutely unnecessary and moreover it is completely unprofessional image showing gating strategy that doesn't really fit into the review.
Figure 1 has been deleted
Line 83-91 is with no reference, although it contains factual information.
As suggested, the references have been completed
Line 92-103 - the first and fifth points do not belong to the mechanisms of NK cell activity, but rather to characterization. I recommend excluding these points.
The last passage about NK cells is very confusing and the section jumps from function thought characterization, education, role in diseases memory and so on. It is really hard to read and not to be lost. The description has to have order, and the paragraph has to follow each other.
In the proposed chapter, we have made changes and introduced a structured description. We hope that the amendments have made the text more readable
Line 222 – the newest database is from December 2020!
Thank you for your suggestion. Of course, the information about the database has been changed to the current one.
Chapter 4 – Typing methods – I really do not understand the inclusion of this section and the detailed description of methods on very low level of scientific language. The end of this section (last paragraph) does not follow up very much, and it is unnecessary.
According to your guidelines, we have deleted the last paragraph. When writing about the methods here, we wanted to show the reader only basic information about the KIR receptor typing methods. Chapter about typing methods has been edited in accordance with the 3 reviewer's guidelines. He suggested additional information, details in the table, changes in the drawings and it had been realized. We hope that the introduced amendments now will please you.
Chapter 5 – sibling donors are not defined as the best donors. There are many criteria for the definition and selection of the best donor and this is not the one of them. The siblings can be haploidentical and therefore, the looking for HLA-matched donors can begin even in case of the existence of siblings. The selection of donors is complex process and is described in the review primitively.
In Chapter 5, we have provided a more detailed description of donor selection. In the review, we wanted to show you the basics as the process is complex. Every day we select donors for our patients. Hence, we are aware of how complex the procedure is. Writing that a fully compatible family donor is the best option, we presented the current position of the EBMT Homebook 2019 Chapter 12.(„ An HLA-identical sibling donor is generally considered the best donor for allo-HSCT; however less than a third of patients will have one available”) As members of the EBMT, we follow these guidelines. I would like to add that the giver of the family is the best cooperative and available. Such a donor cares about a patient. Of course, it is necessary to treat each patient individually. Especially during a pandemic, when unrelated donors are disqualified or fear hospital stays. Then we go beyond the standards, guided by our own experience and epidemiological possibilities. We hope you will be satisfied with the changes.
Followed description of NK cells function are with no references and proper explanation what really happening after HSCT. The figure 6 is very similar to figure 2 and it is unnecessary to repeat the information (very basic and known information) in two sections .
The bibliography was completed and the text was redacted. Figure 6 shows the scenarios of what may happen after HSCT with the NK cell division.
Chapter six can be shortened, and the individual chapters may be common. It would be a good idea to shorten the introduction to reactions that affect KIR receptors and NK cells with a focus only on the relevant topic.
According to your review, Chapter 6 has been re-edited and shortened. There are no separate subchapters, there is one common one in which we focused on the KIR receptors and NK cells in solid organ transplantation
The last chapter about cellular therapy is focused mostly on NK cells and not fully focuses on KIRs. There are several preclinical and clinical studies using blocking of KIR molecules and they have to be described here.
The therapy chapter is supplemented with information on clinical studies using blocking of KIR molecules
Reviewer 3 Report
The review entitled 'KIR receptors as key regulators of NK cells activity in health and disease
’ by DÄ™bska-Zielkowska et al describes the roles of KIR Receptors, which are key regulators of NK cells. The review article is very intriguing, and embodies latest advancements in the field. The section related to transplantation is very interesting and describes the potential therapeutic potential of KIR. The authors could improve the review article.
The article needs thorough English language editing, there are multiple sentences that are unclear.
Other comments
- The authors nicely described KIR in Section 3, It would be nice to have a signalling picture along with the ligands and receptors that are depicted. The downstream regulators are very important and their biological significance.
- In the section KIR Typing methods –
Typing KIR genes- unclear statement
Try to start as ‘Identification of KIR genes’
- It is better to mention the two different methods Flow based and PCR based methods in the initial description in Section 4.
- Table 1, they can add another section and describe if they are quantitative or qualitative and add some more details, needs formatting.
- The method described in section 4, says SSO PCR based method uses hybridization, probes binding, amplification, and detection by luminex it is unclear, can the authors quote relevant references for it. The whole section is without references.
- Figure 5, please add details such as oligo probes, biotin labelled etc., read through luminex as described in the text.
- In Section, relevant references are missing. Can it be depicted as table with the endpoints, response rates, survival rates etc?
- In Abstract- To a high extend? Spelling error
- In Abstract- Which implies coupling of receptors. It can be written in a better way. Binding of ligands (such as MHC I) to the KIR receptors….
- Introduction – alien or foreign particles?
- Section 2- transplanted allogenic tissues or solid organs? Please give some examples
Author Response
The review entitled 'KIR receptors as key regulators of NK cells activity in health and disease’ by DÄ™bska-Zielkowska et al describes the roles of KIR Receptors, which are key regulators of NK cells. The review article is very intriguing, and embodies latest advancements in the field. The section related to transplantation is very interesting and describes the potential therapeutic potential of KIR. The authors could improve the review article.
The article needs thorough English language editing, there are multiple sentences that are unclear.
Other comments
- The authors nicely described KIR in Section 3, It would be nice to have a signalling picture along with the ligands and receptors that are depicted. The downstream regulators are very important and their biological significance.
As suggested, we have prepared the appropriate drawings. We hope it will strengthen and improve our article.
- In the section KIR Typing methods –
Typing KIR genes- unclear statement
Try to start as ‘Identification of KIR genes’
As suggested, we have made changes
- It is better to mention the two different methods Flow based and PCR based methods in the initial description in Section 4.
As suggested, we have made changes
- Table 1, they can add another section and describe if they are quantitative or probes and add some more details, needs formatting.
Table 1 was formatted, we added more information.
- The method described in section 4, says SSO PCR based method uses hybridization, probes binding, amplification, and detection by luminex it is unclear, can the authors quote relevant references for it. The whole section is without references.
References throughout the chapter have been completed
- Figure 5, please add details such as oligo probes, biotin labelled etc., read through luminex as described in the text.
The changes you suggested have been made in the drawing and modificated
- In Section, relevant references are missing. Can it be depicted as table with the endpoints, response rates, survival rates etc?
This point of review is unclear. Would you be so kind and explain more about what you would expect? Which section is this suggestion for?
- In Abstract- To a high extend? Spelling error
The sentence has been modified. Thank you.
- In Abstract- Which implies coupling of receptors. It can be written in a better way. Binding of ligands (such as MHC I) to the KIR receptors….
As suggested, the sentence has been modified, thank you very much.
- Introduction – alien or foreign particles?
As suggested, the sentence was modified to use foreign, thank you.
- Section 2- transplanted allogenic tissues or solid organs? Please give some examples
We have added references where the topic of NK cells in the process of rejection / destruction after transplantation is discussed
Round 2
Reviewer 2 Report
Authors replied to all comments and reshaped the manuscript to be more readable and with correct information. The prolonged chapter about therapy brings very nice demonstration of the importance of the study of KIR receptors and it closes very nicely the review.
The review was very improved and provides readers with complete information on the importance of KIR receptors. I am satisfied with the authors´ answers. Therefore, I have no other comments.
Reviewer 3 Report
The review article entitled 'KIR receptors as key regulators of NK cells activity in health 2 and disease' is interesting. The authors revised manuscript is satisfactory.
This manuscript is a resubmission of an earlier submission. The following is a list of the peer review reports and author responses from that submission.
Round 1
Reviewer 1 Report
The authors have included pertinent information to explain the role of KIR and NK cells in physiological and pathological processes; however, the manuscript did not include a description of KIR and NK cells in autoimmune diseases.
Additionally, there are some details that need to be addressed:
- Review use of abbreviations (IFN-γ, TNF- α, IL-10)
- Homogenize the references and leave space between the "." and the references.
Reviewer 2 Report
The authors have followed the suggested recommendations and profoundly modified them. Among other things:
1- They have included a section on the role of KIR receptors in solid cancer.
3-Reconsidered the section on the role of KIR in pregnancy complications.
2-Revised and increased the number of bibliographic citations.
I believe the manuscript has improved considerably and can now be accepted.
Reviewer 3 Report
General comment
The resubmitted manuscript has unfortunately not improved to such a degree that I will recommend publication. It has several errors and the authors fail to show expertise on NK cells and KIRs to the reader. The title “KIR receptors as key regulators of NK cells activity in health and disease” does not quite reflect the content. (Not sure the grammar in the title is correct?). I do not believe the authors describe NK function or NK receptors in a convincing manner and some of the included figures seems misplaced (contains errors). The authors should decide whether they write a review on NK history and basic knowledge on NK cells and PCR etc, or on the complex role of KIR receptors on NK function.
For future submissions, a more thorough proofreading would be beneficial for the manuscript: typos (as e.g. Heamatology, Klasa Karre (Klas Kärre), consistent nomenclature (eg IL15 vs IL-15), and remove extra spaces etc. There are still many extra spaces etc throughout the manuscript even though commented on in the first round of review.
Abstract: Starting with a confusing sentence. What do the authors mean by “NK cells are part of a component of the cellular immune response”. What component would this be? Delete?
Yes, KIRs are of one of the most important families of receptors, and they are especially important in a transplantation setting. However, the cytotoxic activity of NK cells is not determined by KIR receptors alone. NK cells have a large array of receptors which signals are deciding NK function and fate of target cells, so the statement in sentence 3 is very misleading.
Figure 1 is a basic flow cytometry analysis to define NK cells that would be great for an educational textbook, but don’t think it belongs in this review. The title indicates this is for readers that have some knowledge about NK cells beforehand.
The subheading 2 seems to lack “cells”.
Line 64: There are two main populations of NK cells
Line 66 and 338: Should be IFNg, not INFg
The authors list 5 points of mechanisms for NK cell reactivity (?). This is a confusing mix of listing NK input and function with no clear purpose. Here, they again only consider MHC-binding receptors (KIR and CD92/NKG2 family of receptors). The cytokine-producing function of NK cells are not part of the list but described in the text below.
Klas Kärre is spelled wrong.
Line 114: Not only KIRs bind MHC.
Figure 2 is a simplified and outdated cartoon that has no place in a current review. Even though there is an absence of inhibitory ligands, there has to be interaction for killing to occur.
It has added value to the manuscript that the authors have included a paragraph on NK cell education but I am not convinced the authors are completely updated on the topic. Line 120 makes little sense. One of the referred papers are wrong (Quispe-Tanaya et al, ref.10) but the indicated doi is probably correct, referring to a review by Boudreau and Hsu on NK education. Line 123-124: Something wrong with the sentence? Not sure what the authors mean.
Figure 4 has major errors. Activating KIR receptors do not have intrinsic ITAMs. While inhibitory KIRs have intrinsic ITIM sequences, the activating KIRs interact with the adapter DAP12 containing ITAM sequences. The interaction takes place through a charged amino acid in the KIR transmembrane domain.
Table 1 is messy
Section 4, KIR typing methods, seems a bit misplaced in this review. Basic knowledge of PCR should not be necessary to include.
In figure 6, the authors have introduced an abbreviation “KAR” which is not a current terminology for NK receptors. It is also and not defined anywhere. The use of the term “charge” is misleading in this context. NK function is determined by the balance of activating and inhibitory input.
Section 5 has a misspelled title (correct: “haematology”). This is a very generic headline and should be more specified towards the actual content.
Line 360: Does the author mean “reconstitution” instead of “reconstruction” ?
Line 364: NK cells are not inducing GvL by avoiding GvHD. Maybe the authors mean “while” instead of “by”
Line 372: Lacking “to”
Line 413: …KIR genes….. were selected so they appeared only in the donor before transplantation. Do the authors mean they were only present in the donor? Maybe consider rewriting for clarity
Line 554: Reference should be for first author, not last.
Reviewer 4 Report
The review by J. Debska-Zielkowska et al. reported i) the biology of Natural Killer cells, ii) the description of KIR and corresponding typing methods, iii) the impact of KIR genes and their ligands in hematologic malignancies in the context of HSCT and after solid organ transplantations, iv) KIR/HLA interactions associated with viral infections, after pregnancy complications and solid tumors and v) the role of NK cells in immunotherapies. Although the great interest of NK cells in health and diseases, data from the literature reported in this review are often confusing, paragraphs are not structured, figures need to be improved and some relevant references are missing. Lastly, multiple errors of English spelling and grammar need to be corrected.